# Blood-Vessel-Inspired Hierarchical Trilayer Scaffolds: PCL/Gelatin-Driven Protein Adsorption and Cellular Interaction

**DOI:** 10.3390/polym14112135

**Published:** 2022-05-24

**Authors:** Maria A. Rodriguez-Soto, Andres J. Garcia-Brand, Alejandra Riveros, Natalia A. Suarez, Fidel Serrano, Johann F. Osma, Carolina Muñoz Camargo, Juan C. Cruz, Nestor Sandoval, Juan C. Briceño

**Affiliations:** 1Department of Biomedical Engineering, Universidad de los Andes, Bogotá 111711, Colombia; aj.garcia14@uniandes.edu.co (A.J.G.-B.); ra.riveros11@uniandes.edu.co (A.R.); na.suarez122@uniandes.edu.co (N.A.S.); fe.serrano1899@uniandes.edu.co (F.S.); c.munoz2016@uniandes.edu.co (C.M.C.); jc.cruz@uniandes.edu.co (J.C.C.); jbriceno@uniandes.edu.co (J.C.B.); 2Department of Electrical and Electronic Engineering, Universidad de los Andes, Bogotá 111711, Colombia; jf.osma43@uniandes.edu.co; 3Department of Congenital Heart Disease and Pediatric Cardiovascular Surgery, Fundación CardioInfantil, Bogotá 110131, Colombia; nsandoval@cardioinfantil.org; 4Research Department of Fundación, Cardioinfantil—Instituto de Cardiología, Bogotá 110131, Colombia

**Keywords:** polycaprolactone, oxidation, electrospinning, protein adsorption, hierarchical structures

## Abstract

Fabrication of scaffolds with hierarchical structures exhibiting the blood vessel topological and biochemical features of the native extracellular matrix that maintain long-term patency remains a major challenge. Within this context, scaffold assembly using biodegradable synthetic polymers (BSPs) via electrospinning had led to soft-tissue-resembling microstructures that allow cell infiltration. However, BSPs fail to exhibit the sufficient surface reactivity, limiting protein adsorption and/or cell adhesion and jeopardizing the overall graft performance. Here, we present a methodology for the fabrication of three-layered polycaprolactone (PCL)-based tubular structures with biochemical cues to improve protein adsorption and cell adhesion. For this purpose, PCL was backbone-oxidized (O-PCL) and cast over a photolithography-manufactured microgrooved mold to obtain a bioactive surface as demonstrated using a protein adsorption assay (BSA), Fourier transform infrared spectroscopy (FTIR) and calorimetric analyses. Then, two layers of PCL:gelatin (75:25 and 95:5 *w/w*), obtained using a novel single-desolvation method, were electrospun over the casted O-PCL to mimic a vascular wall with a physicochemical gradient to guide cell adhesion. Furthermore, tensile properties were shown to withstand the physiological mechanical stresses and strains. In vitro characterization, using L929 mouse fibroblasts, demonstrated that the multilayered scaffold is a suitable platform for cell infiltration and proliferation from the innermost to the outermost layer as is needed for vascular wall regeneration. Our work holds promise as a strategy for the low-cost manufacture of next-generation polymer-based hierarchical scaffolds with high bioactivity and resemblance of ECM’s microstructure to accurately guide cell attachment and proliferation.

## 1. Introduction

Cardiovascular diseases represent one of the main causes of death in the world. About 8.5 million people worldwide have diseased blood vessels that must be replaced with vascular grafts [1,2]. Currently, commercially available synthetic vascular grafts are made of several non-biodegradable materials such as polytetrafluoroethylene (PTFE) and polyethylene terephthalate (PET) [3]. However, their performance is inefficient in replacing small diameter vessels (<6 mm) due to their limited long-term patency with only 32% success after 2 years [4,5] caused by thrombogenic events [3], intimal hyperplasia [6,7] and/or atherosclerosis [8], triggering several acute or chronic inflammatory responses [6].

To overcome the limitations of the currently available alternatives, the tissue-engineering field is a strategy for the implantation of a hemodynamic-responsive conduit able to maintain long-term patency [9]. Therefore, biodegradable synthetic polymers (BSPs) are promising platforms for the fabrication of 3D structures [10]. In this sense, tissue regeneration will only proceed via a unique interplay between stem cells and the scaffold, determined by specific 3D architectures with given mechanical properties, and the presence of bioactive molecules [4].

Nevertheless, the lack of surface bioactivity of most BSPs limits their further applicability to promote cell adhesion, growth, and proliferation [6]. For instance, although different tissue engineered vascular grafts (TEVGs) made of BSPs, such as PLA—polylactic acid, PCL—polycaprolactone, PGA—polyglycolic acid, PU—polyurethane, PVA—polyvinvyl acetate, and even PS (polystirene), have been reported to be tested on preclinical animal models, none of them have reached clinical trials as they lack the required features to overcome the current limitations [11,12].

To address this issue, polymers have been functionalized with CO_2_ and NO_2_ plasma to improve hydrophilicity and protein adsorption as well as conjugate bioactive molecules [13] such as adhesive peptides (e.g., CAG, RGD, YIGSR) [14,15] and antithrombogenic agents (e.g., PEG, PEI, heparin) [16,17,18]. Other strategies include blends with natural polymers such as gelatin, chitosan, and cellulose [19]. Despite the successful results in terms of surface bioactivity enhancement, functionalization processes usually involve highly sophisticated and costly procedures and equipment [20] as well as volatile fluorine-toxic solvents (e.g., trifluoroethanol, trifluoracetic acid, hexafluoroisopropanol, among others) to achieve dissolution of blends, which has given rise to several environmental and health concerns due to their end-of-life products [21].

On the other hand, for complex tissues such as blood vessels, esophagus, tracheae, stomach, intestine, bladder and urethra, there is a hierarchical structure comprised of a diverse population of already differentiated cells. Under physiological conditions, the maturation of cells from a single precursor begins with a small population of self-replicating tissue-specific stem cells that will eventually not only differentiate towards a specific lineage, but to a hierarchy of progressively differentiated cells [22]. In the specific case of the cardiovascular system, such a complex structure is given by the native blood vessel ECM, which is composed of three main layers with different features and a composition capable of maintaining vascular homeostasis through the development of different cell phenotypes by means of surface bioactivity, topography, and microstructure [23].

These complex interactions explain the reason for preclinical models on TEVGs showing that nearly 14% of them fail in early stages either in the peri-implantation stages or within the first months, and 35% of them report patency loss or need of removal before the beginning of the tissue remodeling. In this sense, although the graft has been successfully implanted, the substrate is not suitable for tissue remodeling either due to thrombogenesis, generation of exacerbated acute inflammatory responses, or due to lack of regeneration, caused, in both cases, due to limited cell infiltration and interaction given by the BSP properties and scaffold microstructure [4].

Scaffold customization to mimic complex tissues and organs with unique 3D and hierarchical organizations is therefore a major challenge for successful tissue repair. This is especially true when there is a natural interface separating different layers within the native tissue where different types of cells (e.g., endothelial and other functional cells) organize themselves in unique 3D patterns [24]. For instance, for blood vessels, in the innermost layer, called intima, the crosstalk of endothelial cells provides antithrombogenicity and control over muscular tone and inflammation [9]. In contrast, the media layer and adventitia have circumferential aligned collagen and elastin microfibrils that guide the orientation of smooth muscle cells (SMC) to provide the necessary mechanical strength to maintain patency under the circulatory pressure changes [25,26].

In this regard, the rational design of next-generation BSP-based vascular substitutes might be based on the structural design of tubular structures incorporating physicochemical gradients of key components [27]. Forming such 3D constructs might be achieved using additive manufacturing techniques including 3D bioprinting, inkjet printing, and electrospinning. When ECM-fibrillar-like constructs are needed, solvent-assisted methods such as electrospinning seem to be the choice mainly due to the fabrication of fibers in nano- or microscale, control of the scaffold porosity and interconnectivity, and high surfaces areas. Those features allow the cellular interaction and infiltration, as well as the mass transfer for the required nutrient supply and oxygen diffusion [11,28]. Electrospinning can also be implemented in conjunction with solvent-casting and particle leaching techniques to control surface topography while avoiding significant fluid leakage [12].

Electrospinning also offers the opportunity to shape the surface topography and tailor the microstructure of scaffolds by combining the fibrillar-like structure with highly interconnected porous hydrogels fabricated using particle leaching techniques [29]. Here, we aimed at developing a scaffold with tunable microarchitecture and physicochemical cues capable of mimicking the morphological, mechanical, and bioactive cues of the hierarchical structure of the blood vessel’s ECM. This was achieved via the rational and sequential assembly of a tri-layer system by combining solvent-casting manufacture for the first layer and electrospinning and particle leaching techniques for the second and third layer, respectively.

To fabricate an intima-like layer with the surface topography and mechanical resistance observed in native arteries, we employed oxidized PCL (as oxygen-containing functional groups (OCFG) have demonstrated increased biochemical reactivity) deposited on a microgrooved mold manufactured via photolithography. Then, the second and third layers (resembling media and adventitia) were PCL:salt (10%:5%) solutions co-electrospun with varying PCL:gelatin weight ratios (75%:25% and 95%:5% *w/w*) over a rotating mandrel. The polymer blends were obtained via a novel single-desolvation-based method. Gelatin was included to provide controlled biodegradability rates and the possibility of conjugating arginine-glysine-aspartic acid (RGD) resemble motifs, which have been proven useful to allow cellular integration and reduce inflammatory responses [24]. Moreover, to introduce topological cues over the layers, crosslinked glutaraldehyde was implemented followed by salt particle leaching. Correct functionalization of polymers was verified via Fourier transfer infrared spectroscopy (FTIR), TGA, and DSC.

Thus, the manufactured trilayer tubular scaffold exhibited a microgrooved continuous innermost layer, followed by a semi-oriented and porous structure intermediate layer, and finally an outermost layer with prominent fibrillar structures. As proof-of-concept of the feasibility of the engineered scaffold as a vascular graft, protein adsorption capacity, fibroblast cell-seeding, and the release of intracellular reactive oxygen species (ROS) were studied after PCL functionalization confirmation.

## 2. Materials and Methods

### 2.1. Materials

Polycaprolactone (PCL, average molecular weight, 80 kDa), chloroform (99.8%), sulfuric acid (H_2_SO_4_, 98%), potassium permanganate (KMnO_4_, 95%), citric acid (C_6_H_8_O_7_, 99.5%), phosphate buffer saline (PBS), thiazolyl blue tetrazolium bromide (MTT), dimethyl sulfoxide (DMSO, 99%), Dulbecco’s modified Eagle’s medium (DMEM), hydrofluoric acid (48%), Formalin (10%) and Triton X-100 were purchased from Sigma-Aldrich (St. Louis, MO, USA). Type B gelatin was obtained from Químicos Campota (Bogotá, Colombia). Vero cells (CCL-81) and L929 murine fibroblasts (CCL-1) were acquired from ATCC^®^. A lactate dehydrogenase (LDH) kit was acquired from Roche (Basel, Switzerland). Fetal bovine serum (FBS) was obtained from Biowest (Riverside, MO, USA) and DAPI (4′,6-diamidino-2-phenylindole) and Alexa Fluor 594TM phalloidin were obtained from Thermo Fisher Scientific (Waltham, MA, USA).

### 2.2. PCL Oxidation

PCL backbone chemical modification was carried out via oxidation with KMnO_4_ to generate oxygen-containing functional groups (OCFG) as described previously by Sabino [30] including some modifications (Figure 1A). PCL and KMnO_4_ in a 1:5 molar ratio were heated at 80 °C in a water bath for 30 min and mixed with a glass rod in the dark every 5 min. The blend was dissolved in chloroform (10% *w/v*) for 2 h at 50 °C and oxidation was allowed for another 12 h. To remove undesired reaction products (i.e., manganese dioxide—MnO_2_) and the remnant KMnO_4_, the oxidized PCL (O-PCL) solution was washed three times using an excess amount of aqueous H_2_SO_4_ 2% *w/v* by vigorously vortexing with intermediate centrifugations at 4000 rpm for 5 min to remove the supernatants. Finally, three additional washes with aqueous C_6_H_8_O_7_ 15% *w/v* were performed as previously described and the yellowish chloroform solution was freeze dried for 24 h to obtain purified O-PCL (Appendix A).

### 2.3. O-PCL Physicochemical Characterization

PCL oxidation was confirmed via identification of an increase in the OCFG with Fourier transform infrared spectroscopy (FTIR) and calorimetric analysis. Infrared spectra were recorded in an Alpha II FTIR Eco-ART (Bruker Optik GmbH, Ettlingen, Germany) from 4000 to 600 cm^−1^ with a spectral resolution of 2 cm^−1^. The broadening of characteristic peaks of PCL were analyzed to identify backbone structural changes, and hidden peaks in the carbonyl region (1760—1694 cm^−1^) were resolved based on three Gaussian-shaped absorption bands to identify changes in H-bonds between adjacent chains and amorphous and crystalline phases [31].

To compare differences on the thermal degradation profiles that may arise from changes in the covalent bonds of the PCL backbone, a thermogravimetric (TGA) analysis was performed from 25 °C to 600 °C using a ramp of 10 °C/min (ASTM E1131) under nitrogen atmosphere (100 mL/min), in a TA Instruments Q600 thermogravimetric analyzer^®^ (New Castle, DE, USA). Moreover, DSC analysis was undertaken (ASTM D3418) in a TA Instruments Q2000 (New Castle, DE, USA) with a nitrogen atmosphere at a flow rate of 300 mL/min from 25 °C to 130 °C with a heating rate of 10 °C/min. To erase previous thermal history, the instrument was held in an isothermal for 5 min. The samples were cooled down to 25 °C at the same heating rate and then heated to 130 °C as described previously. Data from the first cooling and the second heating steps were recorded and plotted for their analysis (Appendix A).

PCL and O-PCL X-ray diffraction (XRD) analysis was taken in a Rigaku Ultima III X-ray diffractomer (Tokio, Japan) in the Bragg-Brentano configuration. Samples with an average contact area of 25 mm^2^ were placed into a bounding grid of 5 mm and exposed to a Bragg diffraction angle (2θ) sweep between 5° and 50° with a 1.5°/seg step using Kα radiation with a copper (Cu) anode of 40 kV and 40 mA. The Debye-Sherrer equation was used to estimate the apparent crystal size of the polymeric structure based on the full-width at half-maximum of the X-ray diffraction line, also known as FWHM, the wavelength of the X-ray used, which was 1.5406 nm, and the angle between the incident ray and the scattering planes. The phase content of each sample was calculated from the XRD profiles via peak deconvolution using the Materials Data Jade 9^®^ Software (Newtown Square, PA, USA) using Gaussian profiles as fitting methods (Appendix A).

#### Biocompatibility

Considering that the oxidation of PCL could induce changes on the PCL cytocompatibility, hemolytic tendency, platelet aggregation, and activation capacity, biocompatibility assays were conducted following the ISO 10993 standard. The hemolysis assay was conducted using freshly drawn O+ human blood (from a healthy donor) collected in ethylenediaminetetraacetic acid (EDTA) tubes after signing an informed consent (Ethical Committee at the Universidad de Los Andes, minute number 928–2018). The anticoagulated blood was washed 5 times with 0.9% *w/v* NaCl solution at 1800 rpm for 5 min and diluted 1:10 in PBS 1X to obtain an initial stock of 4.25 × 10^6^ erythrocytes/µL. Rectangular samples of the material (0.3 cm^2^) were immersed in 100 µL of the erythrocyte solution for 1 h and 37 °C in a 96-well microplate. PBS 1X and 1% Triton X-100 were used as negative and positive controls, respectively. After incubation, the samples were removed, centrifugated at 1000 rpm for 5 min, and the supernatant absorbance was determined at 454 nm.

Platelet aggregation and activation assays were performed using freshly drawn O+ human blood (from a healthy donor) collected in sodium citrate tubes after signing the informed consent. The anticoagulated blood was centrifuged at 1000 rpm for 10 min to obtain platelet-rich plasma (PRP) and 200 µL was exposed to rectangular scaffold samples (0.3 cm^2^) allowing contact for 3 min and 1 h for platelet aggregation and activation assays, respectively. After incubation, supernatants were removed, absorbance was read at 620 nm from the 3 min exposed samples, and adherent platelets from the samples incubated for 1 h were lysed with 1% *v/v* Triton X-100 for 5 min. Then, LDH working solution (LDH Cytotoxicity Detection Kit) was added to the supernatant and the absorbance was recorded at 493 nm. For platelet quantification, PRP was serially diluted ten times from 3.56 × 10^5^ to 6.95 × 10^2^ platelets/µL to build a calibration curve. Then, absorbance from the calibration curve was fit to a linear regression model (R^2^ = 0.99, Appendix A). Finally, the number of platelets were normalized via the contact surface area (i.e., 0.3 cm^2^). Epinephrine, collagen, and adenosine diphosphate were included as controls of high, medium, and low aggregation.

To verify O-PCL cytocompatibility via both metabolic activity (MTT) and membranal integrity (LDH) in epithelial-like cells, Vero cells were included. Briefly, 100 μL of a cell stock (1 × 10^5^ cells/mL) was seeded and allowed to adhere in a 96-well microplate for 24 h (37 °C, 5% CO_2_) in DMEM media (5% FBS, 1% P/S). Cells were then exposed directly to rectangular scaffold samples (0.3 cm^2^) for an additional 24 and 72 h. DMSO (10% *v/v*) and Triton X-100 (1% *v/v*) were used as positive controls for cytotoxicity in the MTT and LDH assays, respectively. For the LDH assay, the scaffold was removed and 100 µL of the cell’s supernatants was transferred to a new 96-well microplate. Then, LDH reagent solution was added according to the manufacturer instructions and the absorbance was recorded at 493 nm. For the MTT assay, cells were exposed to the MTT reagent solution and incubated for 2 h to allow formazan conversion (37 °C, 5% CO_2_). The formed purple crystals were dissolved with DMSO, and the absorbance was read at 595 nm.

### 2.4. Multilayer Scaffold Fabrication

#### 2.4.1. Fabrication of Microgrooved First Layer

Since intima layers of blood vessels typically include a complex microarchitecture, a microgroove mold was fabricated via photolithography (Appendix A) to imprint a homogeneous micropattern on the luminal layer of the scaffold (Figure 1C). The groove width was 30 μm and the ridge height and weight between repetitive prims of 18 μm and 35 μm, respectively. This was according to porcine carotid artery histological measurements (Appendix A). Accordingly, the first layer of the scaffold was prepared by dissolving O-PCL in chloroform (1:10 *w*/*v*) for 1 h before placing it over the microgrooved mold (Appendix A) in a vacuum oven at 25 °C to allow controlled solvent evaporation and the imprinting of the micropattern. The correct imprinting on the resultant casted film was analyzed via topography measurements along the surface with a profilometer in triplicate (DEKTAK 3, Bruker, Billerica, MA, USA) (Appendix A).

#### 2.4.2. Blends Preparation

PCL:gelatin blends at 75:25 and 95:5 *w/w* were prepared using a novel single-desolvation method. Briefly, PCL and gelatin at 10% *w/v* were dissolved separately in chloroform:glacial acetic acid (3:8 *v/v*) and 80% *v/v* aqueous acetic acid at 50 °C under continuous magnetic stirring. The obtained solutions were mixed by incorporating the dissolved PCL dropwise into the gelatin solution (1 mL/min) at 50 °C under vigorous stirring. This was to allow the controlled gelatin agglomeration through a pH increase for the restoration of gelatin charges through the carboxyl groups (-COOH). In addition, a PCL:salt (1:1 *w/w*) blend was prepared with NaCl microsized crystals dispersed in chloroform (1:10 *w/v*) for 24 h at room temperature via high velocity stirring before the addition and dissolution of PCL to achieve a homogeneous solution [32].

#### 2.4.3. Electrospinning

For the fabrication of the hierarchical porous tubular scaffolds, PCL:gelatin at two different ratios (i.e., 75:25 and 95:5 *w/w*) and PCL:salt blends were co-electrospun over a first layer of micropatterned [27] O-PCL wrapped on a 3.2 mm stainless steel cylindrical mandrel to form a second and a third layer. Before the electrospinning, the O-PCL was sealed onto the mandrel via immersion in glacial acetic acid for 5 s with the microgrooved channels aligned to the mandrel. Each blend solution was co-electrospun through a 16-gauge needle at a flow rate of 1 mL/h using a Cole-Parmer 78-8180C (Vernon Hills, IL, USA) syringe pump, with a needle-to-tip distance of 16 cm and continuous mandrel rotation at 300 rpm for 1 h per layer. The voltage was maintained at 6 kV for the second layer (75:25 *w/w*) and 8 kV for the third layer (95:5 *w/w*) to control fiber deposition and obtain a hierarchical fibrillar porosity gradient [27].

After the electrospinning process, the mandrel containing the scaffold was placed in a glutaraldehyde (25% *v/v*) bath and maintained overnight under continuous rocking movements (10 rpm) using a waving shaker (LabCompanion, Seoul, Korea) to allow gelatin crosslinking at room temperature. The crosslinked scaffold was then removed wet from the mandrel and washed five times with aqueous glutamic acid (5% *v/v* supplemented with 1–8% hydrochloride acid) to remove glutaraldehyde excess every 1 h. Then, a salt leaching process was performed by rinsing the scaffold in distilled water for 12 h at room temperature under gentle agitation at 10 rpm using a waving shaker and stored in PBS 1X at 4 °C until further use.

### 2.5. Scanning Electron Microscopy

Scanning electron microscopy (SEM) inspection was performed in a JSM-6490LV^®^ microscope (JEOL USA Inc., Peabody, MA, USA) with a 10 kV accelerating voltage. Before the examination, samples were cryofractured on liquid nitrogen with perpendicular and parallel cuts with respect to their length, fixed onto aluminum plates with carbon tape, and coated with a thin layer of gold using Vacuum Desk IV Denton Vacuum, Moorestown, NJ, USA) apparatus. SEM micrographs were processed with Fiji^®^ and ImageJ^®^ software packages (version 5.2.0, National Institutes of Health, Bethesda, MD, USA) to analyze the average fiber diameter, size distribution, and relative fiber alignment of electrospun mats by quantifying the diameter of 200 randomly selected fibers in three SEM images at 500× magnification. The relative fiber alignment of the electrospun layers was then analyzed with fast Fourier transform (FFT) [33] by translating optical information into the frequency domain and extracting the radial summing of the pixel intensities in oval projection discretized into 100 points aided by the oval profile plugin. The results were normalized with respect to the minimum value in the dataset and plotted in arbitrary units ranging from 0 to 0.15 [29]. Values above 0.065 units together with two larger peaks observed at 90° and 270° were interpreted as aligned fibers [34].

### 2.6. Protein Adsorption Capacity

As PCL oxidation and PCL:gelatin blending were developed here to improve surface bioactivity of the multilayered scaffold, we hypothesized that protein adsorption would be promoted by the OCFG [35]. Therefore, protein adsorption capacity was evaluated by immersing samples of PCL, O-PCL, and the multilayered scaffold in supplemented DMEM cell culture media (5% *v/v* FBS) for 2, 4, 6, 12, and 24 h at 37 °C. After incubation, the samples were transferred to a new 96-well microplate and exposed to 50 μL of aqueous 1% *w/v* SDS for 30 min at 37 °C. Then, the concentration of protein released in the supernatants was quantified using a bicinchoninic acid (BCA) assay kit (Quanti-Pro, Sigma Aldrich). A bovine serum albumin (BSA) standard curve (Appendix A) labeled BSA with rhodamine B (RhB) (1:1 equivalent ratio). Briefly, a working solution of EDC/NHS and RhB was prepared for 15 min at 37 °C to pre-activate the RhB carboxyl-terminal groups before reaction with 1 mg/mL of aqueous BSA for 24 h at room temperature under constant magnetic stirring. Then, the labeled BSA was dialyzed using a Slide-A-Lyzer™ Dialysis Cassette of 7K MWCO (Thermo Scientific™, Waltham, MA, USA) to remove excess RhB. Finally, labeled proteins were conjugated to samples for 24 h at 37 °C, washed five times with PBS 1X, and imaged with an Olympus FV1000 Confocal Microscope (Tokyo, Japan) under a 20× objective and 559 nm laser excitation. Z-stacks were collected at 10× magnification and stacked from 50 adjacent Z-planes with a stack separation of 2 μm. The spatial distribution of the adsorbed proteins was assessed with the aid of Fiji^®^ and ImageJ^®^ software packages.

### 2.7. Secondary Conformational Changes of Adsorbed Proteins

Secondary conformational changes of adsorbed proteins were estimated using Gaussian deconvolution of the amide I band of the FTIR (1600–1700 cm^−1^) spectra (Section 2.3). Briefly, O-PCL, PCL and the multilayer scaffold were incubated with 1 mg/mL of BSA in PBS 1X for 24 h at 37 °C under constant agitation. After incubation, the supernatants were removed, samples were gently washed twice with PBS 1X to remove unbound BSA, followed by collecting the FTIR spectra and subtracting the corresponding baselines.

### 2.8. Protein Adsorption Kinetics and Isotherms

The adsorption kinetics and adsorption isotherms were measured for PCL, O-PCL, and the multilayered scaffold for six FBS concentrations in DMEM media ranging from 30% to 2.5% (*v/v*) as a function of time (0 to 24 h) at 37 °C. After incubation, adsorbed proteins were quantified as described in Section 2.7 and fit to the pseudo-first-order (PFO) and pseudo-second-order (PSO) kinetic models (Appendix A) [36,37]. Experimental data were also fit to the Langmuir and Freundlich models to estimate possible adsorption mechanisms (Appendix A) [33]. In all cases, adsorbed proteins mass was normalized to a sample volume of 25 mm^3^ for the PCL and O-PCL and 384 mm^3^ for the multilayered scaffold. In all cases, the goodness of fit for the evaluated models was assessed with the correlation coefficient R^2^.

### 2.9. Tensile Strength Test

Tubular specimens with a ring deformation area of 2.49 ± 0.03 mm^2^ were tested using a longitudinal tensile test (ISO 7198: 2016) at a strain rate of 50 mm/min until failure in a INSTRON 3367 (Norwood, MA, USA) before and after 24 h of protein adsorption. In addition, tensile properties of thin films of PCL and O-PCL (L= 100 mm, w = 10 mm, t = 0.156 mm) were recorded (ASTM D882) using a crosshead speed of 12.5 mm/min. Sample thicknesses were measured using a 549 Micrometer (Testing Machines Inc., Amityville, NY, USA), and a total of five samples of each specimen were tested.

### 2.10. In Vitro Cell Morphology and Proliferation

Fibroblasts play a key role in tissue regeneration due to the cell matrix deposition, helping cell colonization on the biomaterial, especially in the external layers of the blood vessels. L929 mouse fibroblasts were used to assess the cell proliferation capacity, morphology changes, and cell infiltration on the inner and the external layers of the multilayered scaffold. Cells were seeded on PCL scaffolds (4 mm × 4 mm × 1 mm) in a 24-well microplate at a cell density of 60,000 cell/sample, allowing cell adhesion for 2 h before immersion in DMEM media (10% FBS, 1% P/S). Samples were then cultured for 15 days (37 °C, 5% CO_2_) with daily media replacement.

After incubation, cells were fixed with 10% *v/v* formalin for 30 min and washed three times with PBS 1X. Subsequently, cell membranes were permeated with 0.25% *w/v* Triton X-100 in PBS 1X for 30 min and washed three times with PBS 1X. Finally, the samples were exposed to a working solution of Alexa Fluor 594TM phalloidin (1:400) and DAPI (1:1000) in PBS 1X, incubated at room temperature for 1 h, washed five times with PBS 1X, and immediately imaged with a confocal microscope (Olympus FLU 1000, Tokyo, Japan) using 488 nm and 358 nm laser excitations. Three images at 20× and 60× magnifications were stacked from 50 adjacent Z-planes with a stack separation of 2 μm. Two-dimensional cell cultures (10,000 cells/cm^2^) over a 18 mm glass slide coated with Poly-D-Lysine (PLL; Sigma, P4832, 0.01% aqueous solution) were included as reference for cells morphology and as positive control of the cell culture incubated for 24 h. Cell morphology was analyzed via an aspect ratio (major axis/minor axis) estimation based on 50 cells measured randomly from confocal images collected at 60× aided by Fiji^®^ and ImageJ^®^ software packages. In addition, samples of seeded multilayered scaffolds were cryofractured in liquid nitrogen after cell fixation, air-dried overnight, and imaged via SEM, as described previously.

### 2.11. Cell Viability

A LIVE/DEAD^®^ cell viability assay (Life Technologies, Dublin, Ireland) was used to evaluate cell survival in the multilayered scaffold in both the lumen and external surface of the tubular structure at day 15. Three images from randomly selected regions along three samples were inspected at 20× magnification via confocal microscopy and stacked from 25 adjacent Z-planes as described previously. The cell viability percentage was then calculated using the ratio of live cells to the total number of cells using image examination in Fiji^®^ and ImageJ^®^.

### 2.12. Intracellular Oxidative Stress Levels

Reactive oxygen species (ROS) are involved in several physiological and pathological processes as signaling molecules that mediate specific cellular responses in inflammation and healing phases of diseased organs. Moreover, since ROS release overtime has been reported to be implicated in the orchestration of the inflammation and cellular dysfunction upon the implantation of BSP, we hypothesized that by increasing surface bioactivity through modification with cell adhesion motifs present in gelatin (i.e., RGD), ROS production over time will decrease via an increased protein adsorption. Therefore, to test this hypothesis, intracellular ROS quantification assay was performed using the dihydroethidium (DHE) assay after 1 and 15 days of exposure to THP-1 monocytes (used as representative immune cells) in RPMI (10% FBS) medium. After incubation, cells were exposed to DHE working solution and confocal images at 20× were stacked from 25 adjacent z-planes collected at 559 nm. Finally, fluorescence intensity was analyzed using Fiji^®^ and ImageJ^®^ software packages.

### 2.13. Statistical Analysis

Data were analyzed statistically using Graphpad Prism^®^ 9.1.1 software (Windows, GraphPad Software, San Diego, CA USA, www.graphpad.com, accessed on 10 May 2022) using a two-way ANOVA test with Tukey’s multiple comparison of means after checking for normality, independence of observations, and homoscedasticity. Normally distributed data is presented as mean ± standard deviation, and *p*-values below 0.05 (*p* < 0.05) were considered significant. Grubbs’ test was included to detect single outliers of the data sets (data not shown). Figure 1 shows schematically the workflow for the scaffold fabrication.

## 3. Results

### 3.1. PCL Oxidation

#### 3.1.1. FTIR Spectroscopy

Functionalization of PCL backbone and absence of undesired reaction product remnants was confirmed via FTIR analysis (Figure 2a–f). Characteristic peaks of PCL were identified between 1710 and 1730 cm^−1^ (ester carbonyl -C=O stretching vibration) and 3460 and 3555 cm^−1^ (hydroxyl group -OH stretching vibration) [34,38]. Furthermore, a broadening of –COOH related vibrations was identified at 1719 cm^−1^ (C=O stretching vibration), 2795 cm^−1^ (C-H stretching vibration), and 2900 cm^−1^ (-OH bending) (Figure 2d–f) [31]. The broadening identified in the PCL-O FTIR spectra confirm that oxidation served as a useful and low-cost method to improve PCL chemical reactivity, which was critical to allow its covalent surface functionalization to promote protein adsorption and consequently improved cell adhesion. Figure 2j shows the possible products of PCL oxidation to obtain OCFG as described by Sabino et al. [30].

Given the semicrystalline structure of PCL, apparent changes in carbonyl terminals were studied to estimate possible changes in chains organization on the basis of three Gaussian-shaped absorption bands (Figure 2a,b) [31,39]. The maximum absorbance for the deconvoluted peaks was fixed at 1735, 1719, and 1698 cm^−1^, since they have been reported to be related to the amorphous and crystalline phases and hydrogen bonds between polymeric chains in PCL [31]. The results of normalized intensities are shown in Table 1. An increase of 19.5% in the C=O indicates a transition of the carbonyl groups to the amorphous phase in the PCL-O, which is predominant in the remanent crystalline phase. In addition, the hydrogen-bonded carbonyl vibration seemed to increase by about 17% in O-PCL, which can be explained by an increase in the h-bonds between adjacent chains. This provides further evidence regarding the increase in chemical reactivity upon PCL oxidation.

#### 3.1.2. Thermogravimetric Analysis

TGA was performed to determine differences in the thermal degradation profile of the O-PCL when compared to the pristine PCL. Thermal degradation of pristine PCL was generated in a single step of weight loss with an onset at 400 °C as shown by the DTGA curve extrapolation of the slope connecting the maximum peak to the baseline of the thermal profile (Figure 2g,h). Upon PCL oxidation, an early sharp degradation takes place in an additional weight loss step, with an onset temperature of 320 °C that can be attributed to the decomposition of OCFG [40]. Thus, the first weight loss of the O-PCL, resulted in an increase of 18% in the OCFG most likely due to additional covalent bond formation [41], which is in agreement with the transition of amorphous C=O groups estimated by the FTIR analysis.

Changes in the DSC thermogram of the PCL and the O-PCL are shown in Figure 2i and Table 2. Crystallization and melting behavior of pristine PCL was found to be consistent with the DSC scans reported previously [30,42,43]. Upon oxidation, the crystallization exotherm and melting endotherm peaks shifted by about 12% and 13%, respectively. These results provide further support to the TGA analyses as it has been reported that the presence of OCFG results in less stable covalent bonds that can be broken down by temperature to achieve melting [41,44]. Moreover, the presence of OCFG interrupts the linear crystallizable sequences, forcing rearrangement of the semicrystalline structures into amorphous lamellar crystals as evidenced by the 12% reduction in the crystallinity degree (Xc) [30]. This was confirmed via X-ray diffraction (XRD) measurements (Appendix A).

#### 3.1.3. X-ray Diffraction

XRD pattern of PCL and O-PCL films at room temperature displays three strong diffraction peaks at Bragg angles 2θ = 21.4°, 22°, and 23.8° that correspond to the orthorhombic planes (110), (111), and (200) of the crystalline phase of PCL. Furthermore, a halo centered near 21° in the XRD profile indicates the presence of amorphous structures in the samples as expected from the semicrystalline structure of PCL. Observable differences can be identified in the deconvoluted peak intensities and peak widths among the orthorhombic unit cell diffractions when oxidation takes place, indicating variations on the lamellar organization of the polymer.

Nevertheless, since there were no additional peaks on the XRD pattern of the O-PCL, it can be inferred that the oxidation process did not significantly affect the semicrystalline structure and, thus, only appears to be a surface phenomenon [3]. Finally, the XRD pattern is comparable with those obtained in previous studies where OCFG on the backbone of PCL was generated with CO_2_ plasma treatment, indicating that the oxidation introduced here is a cost-effective alternative for polyester oxidation (Appendix A).

#### 3.1.4. O-PCL Biocompatibility

The hemolysis percentage of the O-PCL (Figure 3a) remained below 1% when compared to pristine PCL. Platelet aggregation triggered by the O-PCL reached 17.07 ± 7.05% (Figure 3b) and the presence of adherent platelets increased by 3.57% after 1 h (Figure 3c). However, there was no statistically significant difference (*p* < 0.05) when compared with PCL. Vero cells remained metabolically active with no major changes in cell membrane integrity after 24 and 72 h as demonstrated by cell viability levels above 95% for both MTT and LDH assays (Figure 4d–f). Furthermore, no apparent cell morphological changes were observable. These biocompatibility results indicate that O-PCL fulfills the criteria of the ISO-10993 standard, which is critical to move forward with in-vivo studies in the next stages.

### 3.2. Scaffold Characterization

#### 3.2.1. Protein Adsorption Capacity

Protein adsorption has a key role in the first stages right after biomaterial implantation and particularly as the main initial process prior to cell adhesion on polymeric biomaterials such as PCL. In this regard, here we measured changes in the FBS protein adsorption profiles by quantifying the proteins detached after SDS treatment via a BCA assay (Figure 4a). Protein adsorption increased by about 86% and 91% in O-PCL and the multilayered scaffold after oxidation and gelatin blending (which contains abundant carboxyl and amine groups), respectively. These results indicate an increase of approximately 1.8 μg and 3.5 μg in the adsorption of serum proteins per cm^2^ after 24 h for O-PCL and the multilayered scaffold when compared to the pristine PCL. In addition, adsorption kinetics experiments showed that at least 12 h are required to saturate the surface-active functional groups of the samples (Figure 4a) [33]. Furthermore, a confocal microscopy Z-stacking reconstruction (Figure 5) showed the presence of protein clusters along the multilayer scaffold, possibly due to preferred adsorption over patches of highly crosslinked gelatin regions. This observation provides further evidence for the notion that by including surface polar functional groups, it is possible to enhance surface—protein interactions. This may be useful to promote cell adhesion over the innermost and outermost layers of the scaffold [36].

PCL oxidation seems to be an efficient backbone modification for promoting protein-surface interactions since SEM inspection revealed the presence of highly packed nanoclusters after 24 h of protein adsorption. In contrast, protein adsorption on PCL led to randomly distributed protein agglomerates (Figure 4). This behavior seems to be consistent with previously reported micrographs where protein adsorption mediated via electrostatic interactions leads to highly ordered nanostructures on the surface that might be useful to improve cell adhesion [45,46].

Without any treatment, BSA exhibits five characteristic absorbance peaks that relate to antiparallel β-sheets (1680 cm^−1^), α-helices (1650 cm^−1^), parallel β-sheets (1630 cm^−1^), and intermolecular interactions between secondary structures (1610 cm^−1^) [47]. After 24 h of exposure to the PCL-based materials, BSA undergoes adsorption-induced conformational changes as evidenced by an initial loss of the α-helical content (Table 3). This implies that upon contact with the material’s surface, BSA partially transitions to less ordered secondary structures such as extended β-sheets.

The highest degree of unfolding was found for neat PCL, where the α-helical components disappeared and were replaced by β-sheets and even completely disordered random coils (Table 3). This result strongly suggests that adsorbed proteins are likely to have lost their 3D structures (Figure 4) and, consequently, fail to properly bind to the ligands needed for cells adhesion and secondary protein aggregation [45,46]. In contrast, the multilayer scaffold modified with gelatin and the oxidized PCL seem to prevent the drastic secondary structural changes observed for pristine PCL and most likely caused by its higher hydrophobicity compared with the modified polymers.

As BSA adsorbs and exposes residues typically buried in the native structure, secondary protein adsorption appears to be promoted, which includes diluted species such as fibronectin (0.07% of FBS) and vitronectin (0.11% of FBS) [45]. This, in turn, appears to lead to proteins transitioning into ‘active’ conformations by exposing RGD adhesion motifs, which have been thought to promote cell adhesion (Figure 4g), as has been reported previously by Koblinski et al. [48] and Thamma et al. [49].

#### 3.2.2. Protein Adsorption Kinetics and Isotherms

The adsorption mechanisms of FBS proteins on the PCL-based materials was studied with the aid of the Langmuir and Freundlich models by linearizing the adsorption capacity of the adsorbent surface (*q_e_*) as a function of the adsorbate’s concentration (i.e., BSA) at equilibrium (Appendix A). The parameters recovered for the two models under consideration are shown in Table 4. Protein adsorption data on the O-PCL and the multilayer scaffold were fit best by the Freundlich model, while the PCL data were fit best by the Langmuir model, as evidenced by the obtained correlation coefficients [50]. This suggests that adsorption on the O-PCL and the multilayered scaffold is likely to occur by forming an uneven multilayer, while that of pristine PCL proceeds through a homogenous monolayer. This result provides further evidence for the notion that BSA adsorption is largely responsible for recognition of diluted protein species that trigger multilayer deposition in the samples where the protein secondary structure remains ordered (i.e., high α-helical contents) (Figure 4g). The recovered models’ parameters (*R_L_* < 1 and *N* > 1) suggest that adsorption prevails over desorption (i.e., protein release) for the O-PCL. This is corroborated by the higher values of *K_f_* and *K_L_* after 24 h (Table 4) [51]. This can be attributed to the presence of positively charged residues (e.g., lysine, histidine) on BSA’s structure, which can interact easily with negatively charged functional groups such as the OCFG [52].

The adsorption kinetics data (Appendix A) were fit to the PSO and PFO models with a higher agreement obtained for the former (Table 5) along with a higher accuracy level as suggested by the smaller values for the mean absolute error (MAE) and root mean square deviation (RMSE). In addition, the equilibrium constant (*K_2_*) reveals an inverse dependency with the sample’s hydrophilicity, since it was found that it decreases for samples with functionalized surfaces. This has been reported previously for OCFG functionalized biomaterials [53].

#### 3.2.3. Micropatterning and Fiber Size and Alignment

A close inspection of SEM micrographs (Figure 6) shows evidence of a normal distribution of electrospun microfiber mats with an average fiber diameter of 4.61 ± 1.07 μm (Figure 6i) in the external layer of the scaffold (i.e., 95:5 PCL/Gelatin). SEM micrographs were also analyzed via FFT (Figure 6j,k) to quantitatively determine the alignment orientation of PCL microfibers within the mats. The results revealed the prevalence of vertical fiber alignment (radial direction) as can be deduced from the maximum value of the 2D FTT (~0.14) and the prominent peaks at 90° and 270°. However, the presence of sharp peaks at 180° suggests that there are also some horizontally aligned fibers mats. The existence of these two fiber orientations indicates that the formed structure exhibits a certain level of anisotropy most likely due to the collector rotating velocity [27].

Similar observations of partially aligned mat fibers were made by Vaz et al. [27] and Wu et al. [54] where vertically (90° to 270°) and horizontally (180° to 360°) aligned fibers were obtained when the mandrel rotating speed was maintained below 1000 RPM, which additionally led to fiber diameters in the nanoscale range. The differences in fiber alignments could also be explained by the significantly different viscosities of the co-electrospun solutions, which in one case tend to deposit vertically oriented fibers and in the other horizontally [32,55]. Despite such differences, the electrospun fiber exhibited an approximately uniform diameter distribution that allowed sufficient scaffold interconnectivity as has been reported for diameters above 4 µm. This feature is critical for improving cell infiltration and attachment [56].

### 3.3. Tensile Properties

The maximum strain (i.e., tensile elongation), elongation at break, and Young’s Modulus were taken from the failure samples and are shown in Table 6. As expected, solvent-casted samples (i.e., PCL and O-PCL) exhibited superior resistance to uniaxial tensile loading, where a maximum in tensile performance was observed for PCL [57,58]. The reduction in the tensile properties for the electrospun scaffold compared to the ones based on PCL could be attributed to the presence of gelatin, which has a much lower tensile resistance than commonly used BSP [59,60]. Moreover, the reduction in the tensile strength and elongation at break, along with the increase in the Young’s modulus for the O-PCL, likely indicate a transition to a brittle material [61]. This observation could be rationalized by a reduction in crystallinity, which in turn is likely to be caused by the oxidation process as evidenced by the tight organization of the lamellar regions, which allows an efficient energy transfer capacity but impedes energy accumulation [62].

The tensile measurements after 24 h of FBS adsorption confirmed an increase in stiffness for both O-PCL and the multilayer scaffold; however, the PCL remains practically unchanged. These changes after protein adsorption could be attributed to an increase in the ionic interactions between the pendant functional groups of polymeric chains and the deposited protein layer, which restricts chain mobility and energy transfer capacity. Interestingly, this also correlates well with the observed initial increase in the mechanical performance as indicated by the Young’s modulus and, later, the early failure caused by the emergence of nucleation points (tensile strength and elongation) [63]. The presence of such nucleation points has been correlated with an increase in the chain length and, consequently, with an increase in energy transfer capacity during elastic deformation. Furthermore, this favors an early and fast energy dissipation during plastic deformation via the breakage of covalent bonds [61]. Moreover, there seems to be a correlation between the adsorbed protein level and the extent of changes in the tensile strength.

### 3.4. Cell Attachment Performance In Vitro

After 15 days of culture, L929 mouse fibroblasts were found successfully attached to the multilayered scaffold, as well as on the innermost O-PCL layer (Figure 7a–c). Moreover, examination of the cross-sectional cryofractured and F-actin/DAPI stained cultured scaffold reveals preferential attachment of cells to the gelatin-rich porous regions (Figure 7c) with filopodia extending over the material (Figure 7a–c) [61]. This preferential cell adhesion seems to be consistent with similar observations reported previously for fibroblasts seeded on gelatin and collagen-containing nano- and micro-fibers. Moreover, fluorescence images (Figure 7a,b) after 15 days of cell culture showed elongated morphologies that are typically observed for active fibroblasts. Finally, cluster-based proliferation was observed with a marked presence of 3D filopodia elongation as evidenced by an aspect ratio greater than 2 [64].

LIVE/DEAD^®^ cell staining showed high cell viability (~80%) after 15 days of culture for the fibroblast population seeded on the outermost surface of the scaffold (Figure 7c). Similar viability results were found (above 80%) if the O-PCL’s innermost layer was seeded instead. These findings indicate that the porous structure of the electrospun PCL:gelatin mesh and the casted O-PCL are well suited for cell proliferation by providing the macro- and micro-porosity for nutrient perfusion and cell attachment and filopodia extension [65].

The cell nuclei distribution profile recovered from Z-stacks (Figure 7f) demonstrates successful fibroblast distribution along the z-axis with a favored adhesion to the second layer (75:25 *w/w*), which locates between approximately 0 and 50 µm radially. These results provide further evidence for the preferential cell adhesion to the RGD motifs of gelatin [66], which were lightly crosslinked to favor fusion of electrospun fibers predominantly along the media layer (Figure 7C). Cells were found also attached to the fibers themselves (Figure 7C).

THP-1 intracellular ROS quantification using DHE mean fluorescence intensity in day 15 remained at about the same level as day 1 with statistical significance (*p* < 0.05). These results suggest that ROS production via monocytic cell infiltration on the scaffold is negligible, and therefore, it is very likely that in vitro acute inflammation can be avoided [67,68]. These findings agree well with previous studies where ECM-like morphological structures (i.e., microfibrils) and peptide-resembling motifs have been introduced using gelatin-rich regions and electrospinning manufacturing [69].

## 4. Discussion

Cell behavior is primarily mediated by the chemo-mechanical sensing of its surroundings. Therefore, controlling the physicochemical cues of implantable porous polymeric scaffolds is key to guide cell infiltration, adhesion, and proliferation. Moreover, they define the interactions of macromolecules (e.g., serum proteins) during early stages of regeneration to promote adequate bioactive surfaces that also allow correct mechanical responses. Additionally, the protein adsorption profile on polymeric materials has been reported to reduce short-term inflammation mediated by the production of ROS [11,69,70]. Conversely, tailoring the surface bioactivity of BSP to improve serum protein adsorption while avoiding major conformational changes seems to be a promising approach to improve tissue resemblance and guide cellular response. In this study, we developed a low-cost and highly sophisticated functionalization of a multilayered scaffold based on PCL that contains fundamental microstructural features that resemble the hierarchical organization of blood vessels’ ECM. This was achieved by combining ECM-like microfibrils obtained via electrospinning and particle-leaching techniques [71].

The functionalization strategies implemented here are highly advantageous because they are relatively inexpensive and involve simple protocols. Biochemical cues were incorporated by creating a gradient of gelatin within the matrix, which besides high availability and low cost takes advantage of the presence of peptide-resembling motifs (i.e., RGD) and amine groups [71]. Furthermore, the proposed PCL:gelatin blending strategy represents an alternative method that is free of fluorine-based solvents, contrary to the commonly used techniques for blending, which have been criticized for their environmental impacts and safety issues [72].

Moreover, the blending of PCL:gelatin helps to maintain the shape fidelity of the electrospun nanofibers as demonstrated by the cross-sectional and surface SEM imaging of cryofractured samples, even after the glutaraldehyde-mediated crosslinking of gelatin. Additionally, the combination of interconnected nanofiber patches with highly porous gelatin microstructures produced by salt leaching leads to the presence of bulk interconnected pores and mats that allow the efficient nutrient, oxygen, and macromolecule exchange required for cell survival (Figure 7) [26,34]. This was evidenced by the imaged cells infiltrated and attached to the scaffolds, the protein adsorption patterns (Figure 6), and the throughout cell nuclei distribution (Figure 7f). Successful manufacture of the physicochemical gradient was confirmed by the high abundance of fibrillar-like structures along the outermost layer and the high porosity of the intermediate one where cells preferentially attach (Figure 7f), possibly due to a differential protein adsorption profile. These results provide robust evidence of the hierarchical organization formed in the trilayered scaffold where cells can attach and proliferate effectively.

The circumferentially aligned mesh demonstrated via the FFT analysis (Figure 6) revealed anisotropic orientation, which is ideal for mimicking the blood vessels’ ECM. This unique feature will be useful to allow a graduate alignment of smooth muscle cells (SMCs) [73], which are responsible for supporting the mechanical stress induced by blood flow and providing the necessary compliance [73,74,75]. Such needs will also be addressed by the high elasticity and mechanical resistance resulting from the combination of O-PCL, PCL, and gelatin (Table 6. These features make the fabricated multilayer scaffold optimal for bearing load applications (i.e., active structural element) that could withstand the fluid dynamics of blood vessels [76].

## 5. Conclusions

One of the major challenges of manufacturing vascular grafts is to assure that they not only have geometries that are like those of native vessels but that they mimic their tissue organization and chemistries. This is critical to promote the appropriate biochemical and mechanotransduction signals that lead to controlled cell-surface interactions. The present study was dedicated to addressing this challenge by tuning the topological and bioactive cues of hierarchical tubular structures using a layer-by-layer rational design to assure control over protein adsorption capacity and the subsequent cell adhesion events. This was achieved by incorporating a radial gelatin gradient into the manufactured multilayer scaffold, which was composed of electrospun polycaprolactone (PCL) microfibrils deposited on a microgrooved oxidized-PCL (O-PCL) film. The obtained scaffold showed interconnected pores, which were formed after particle leaching conducted on the PCL microfibril structure. The achieved 3D topology mimicked very closely that of the media and adventitia layers of the blood vessels’ ECM as evidenced by the preferential cell attachment and protein adsorption observed along the intermediate layer. Our PCL-based multilayer scaffold also proved to be a suitable platform for fibroblast cell attachment and proliferation where cells exhibited elongated and spindle-like morphologies, which are typically observed under native conditions. Additionally, the scaffold maintains the basal intracellular ROS production levels of immune cells (THP-1).

All in all, our work highlights a relatively simple and cost-effective route to engineer multilayered tubular scaffolds with surface and bulk features that resemble native ECMs. We are certain that this is one step forward to approaching hierarchical structures that provide the physicochemical and mechanical cues to overcome early failure of chemically inert BSP. Future work will focus on studying the interplay of cell phenotypes at the engineered vascular wall to identify possible design improvements to elicit a desired cellular response, attachment and ROS signaling profile.

## 6. Patents

Collagen based multilayered regenerative vascular graft with bioactive luminal coating. (Original in Spanish: *Injerto vascular regenerativo multicapa basado en colágeno, con recubrimiento luminal bioactivo*). Patent application number in Colombia: NC2021/0017700, submitted on 22 December 2021. 

## Figures and Tables

**Figure 1 polymers-14-02135-f001:**
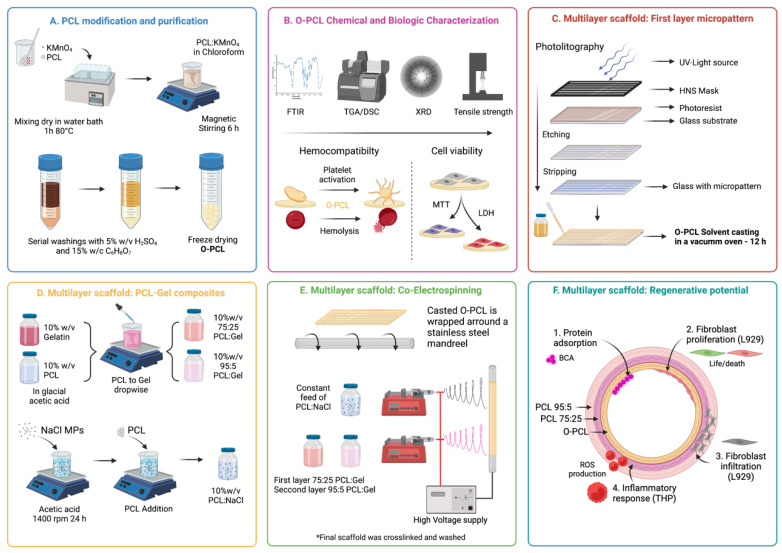
Schematic representation of the workflow for the fabrication of a tri-layer PCL-based scaffold with a physicochemical gradient of gelatin. (**A**) PCL oxidation with KMnO_4_. (**B**) Functionalization confirmation and biocompatibility assessment. (**C**) First layer fabrication by imprinting on a microgrooved mold manufactured via photolithography. (**D**) PCL:gelatin blending via a novel single-desolvation-based method. (**E**) Electrospinning of PCL:gelatin blends for the manufacture of the second and third layers. (**F**) Cellular and bioactive response of the multilayer scaffold via protein estimating adsorption capacity, cell viability, F-Actin staining and intracellular ROS levels.

**Figure 2 polymers-14-02135-f002:**
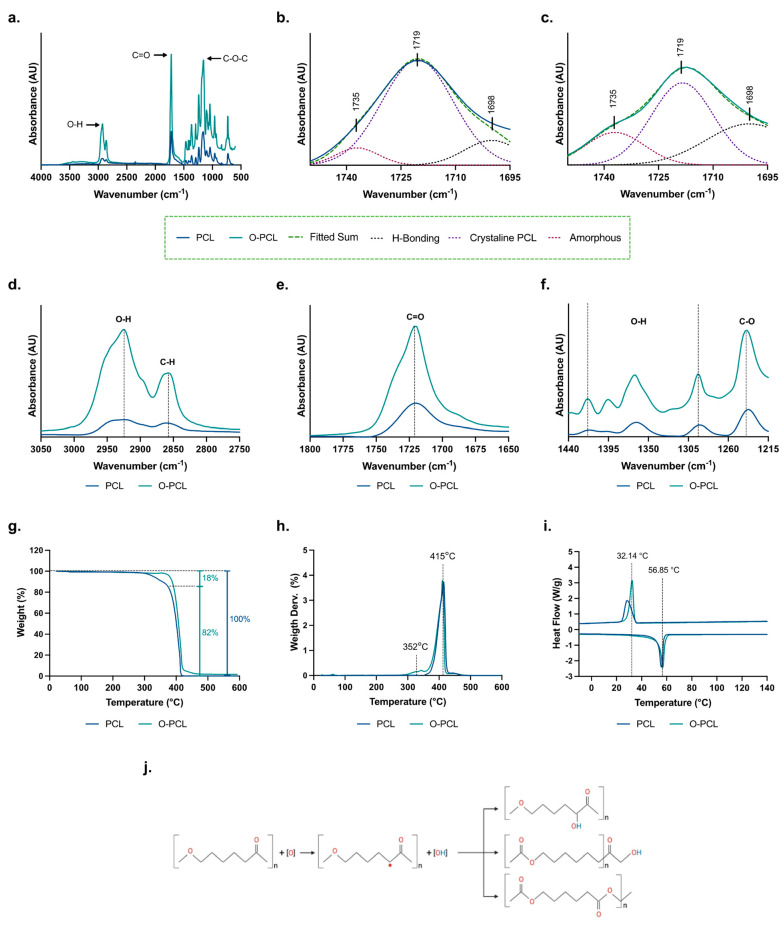
Evidence of PCL oxidation. (**a**) FTIR spectra of PCL and O-PCL measured in the range between 4000 to 600 cm^−1^. Deconvolution analysis of carbonyl (C=O) stretching via Gaussian profiles of (**b**) PCL and (**c**) O-PCL. Maximum absorbance of the deconvoluted curves is indicated in the figure. AU stands for arbitrary units. Magnification of the (**d**) hydroxyl (-OH) and carbon-hydrogen (-CH) stretching, (**e**) carbonyl (C=O) stretching, and (**f**) carbon-hydrogen (-CH) and carbon-oxygen bending (-CO), present in pristine PCL and the O-PCL because of an increased level of oxygen-containing functional groups. (**g**) Thermogravimetric (TGA) analysis and (**h**) DTGA curves of PCL and O-PCL. (**i**) DSC thermograms for pristine PCL and O-PCL. (**j**) Possible outcomes of the PCL oxidation process.

**Figure 3 polymers-14-02135-f003:**
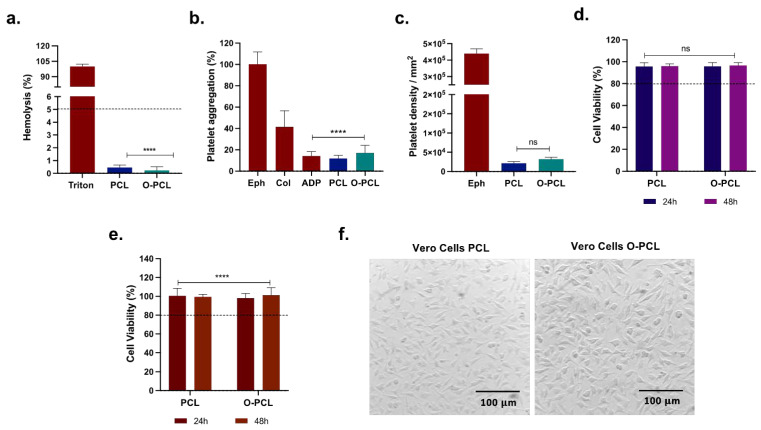
Biocompatibility assessment of the O-PCL. (**a**) Hemolytic behavior after an incubation time of 1 h. Triton X-100 and PBS were used as positive and negative controls. (**b**) Platelet aggregation assay. Epinephrine (Eph), collagen (Col), and adenosine diphosphate (ADP) were used as high, medium, and low aggregating references, respectively. (**c**) LDH platelet aggregation assay after 1 h of blood plasma exposure. (**d**) MTT and (**e**) LDH cytotoxicity in Vero cells after 24 and 72 h of exposure to the materials, (**f**) images of Vero cells seeded over PCL and O-PCL after 72 h of exposure. (ns is *p* > 0.05, **** is *p* ≤ 0.0001).

**Figure 4 polymers-14-02135-f004:**
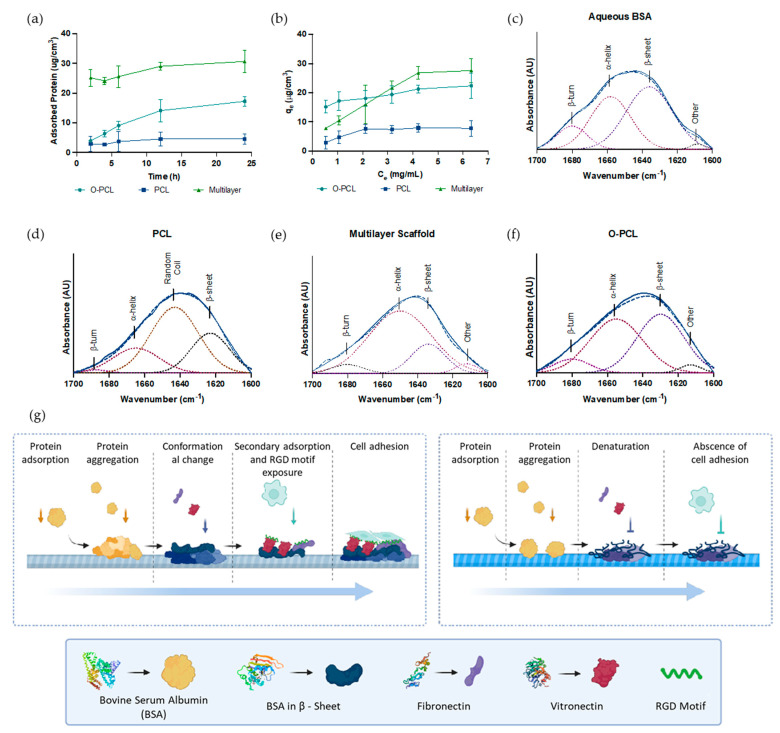
FBS protein adsorption. (**a**) Proteins adsorbed on a per volume basis as quantified via the BCA assay. Almost 12 h were needed to reach a steady state in protein concentration. (**b**) Protein adsorption isotherms. Secondary conformational changes of BSA estimated via FTIR deconvolution of the amide I absorption band (1600—1700 cm^−1^) of (**c**) aqueous BSA (control of native structure) and BSA adsorbed on (**d**) PCL, (**e**) O-PCL, and (**f**) the multilayer scaffold. (**g**) Schematic representation of the proposed adsorption mechanism of FBS proteins based on experimental data fitting to the Langmuir and the Freundlich models.

**Figure 5 polymers-14-02135-f005:**
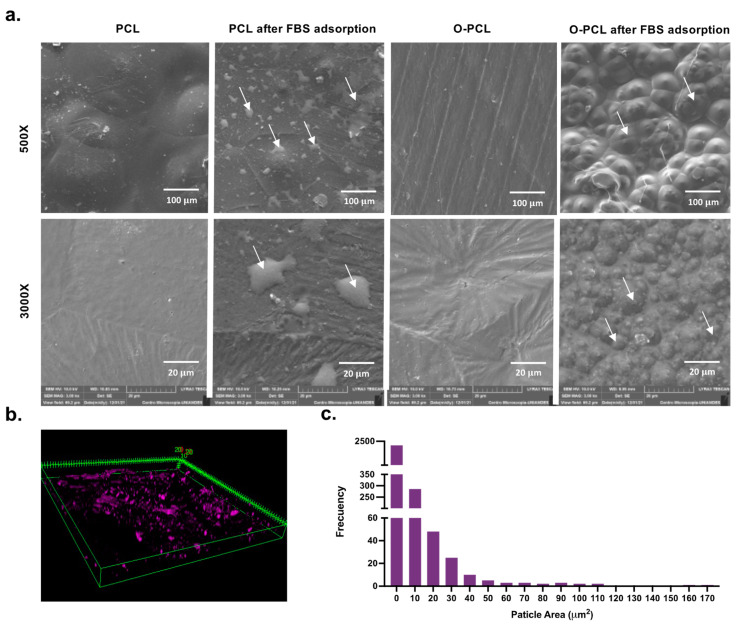
(**a**) SEM surface morphology of PCL and O-PCL before and after 24 h of FBS protein adsorption at 500× and 3000× magnifications. Arrows point to protein aggregation clusters. (**b**) Confocal imaging at 20× magnification of RhB-labeled albumin adsorbed on the multilayer scaffold. Reference dimensions are shown in micrometers (μm). (**c**) Histogram for particle size distribution of the adsorbed proteins as calculated using Fiji^®^ and ImageJ^®^.

**Figure 6 polymers-14-02135-f006:**
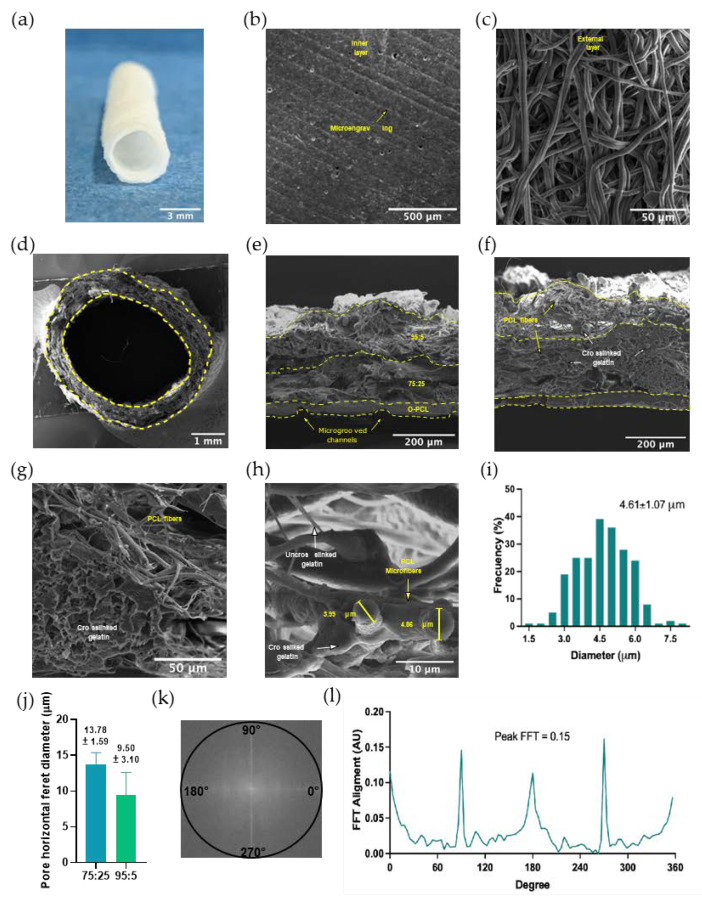
PCL:gelatin hierarchical electrospun tubular scaffolds with an innermost layer of microgrooved oxidized PCL (O-PCL). (**a**) Macroscopic view of tubular scaffold with an internal diameter of about 3.2 mm. (**b**) SEM micrographs of the scaffold’s inner layer showing the microgrooved pattern fabricated via solvent casting and (**c**) of the external 95:5 (PCL:Gelatin) electrospun layer. Cross-sectional SEM visualization of (**d**) a cryofractured tubular PCL-based scaffold. SEM micrograph of a cryofractured tubular scaffold with perpendicular (**e**) and parallel (**f**) views in respect to the scaffold length. (**g**) Magnification of the interface between the 75:25 and 95:5 (PCL:gelatin) layers. (**h**) Representative cross-sectional view of a cryofractured PCL fiber and (**i**) histogram of the fiber diameter distribution. Horizontal ferret diameter of wall pores considered to be the smaller (**j**) ImageJ^®^ FFT frequency plot (**k**) and 2D FFT alignment plot (**l**) of SEM micrographs of electrospun fibers. The 2D FFT peak value and the high-frequency pixels placed away from the origin and toward the periphery of the frequency plot indicate an alignment fiber tendency.

**Figure 7 polymers-14-02135-f007:**
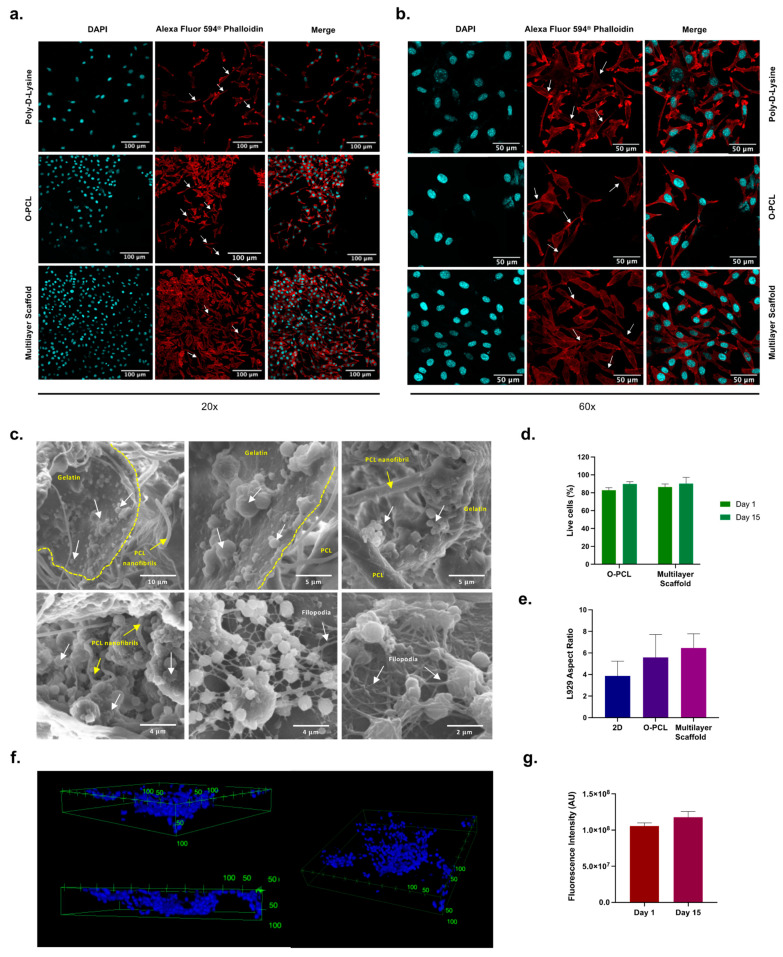
Cellular response upon after 15 days of culture. F-actin and DAPI stained L929 mouse fibroblasts at 20× magnification (**a**) and 60× magnification (**b**). White arrows point to the spindle-like morphology achieved by seeded cells. (**c**) SEM inspection of the cell seeded cryofractured samples. White arrows point to cell clusters and filopodia. (**d**) Live/dead analysis. (**e**) Aspect ratio (AR) of L2929 after 15 days of cell culture. (**f**) Cell nuclei distribution as reconstructed from confocal Z-stacks. Reference dimensions are shown in micrometers. (**g**) DHE fluorescence intensity after 1 and 15 days of THP-1 culture.

**Table 1 polymers-14-02135-t001:** Normalized intensity of the deconvoluted carboxyl absorption band.

Sample	Maximum Absorbance (cm^−1^)	PCL	O-PCL
Amorphous	1735	14.18%	33.73%
Crystaline	1719	78.15%	48.18%
H-bonding	1698	7.67%	18.10%

**Table 2 polymers-14-02135-t002:** DSC data of PCL and O-PCL.

Sample	Cooling Scan		Second Heating Scan		X_c_ (%)
	T_c_ (°C)	∆H_c_ (J/g)	T_m_ (°C)	∆H_m_ (J/g)	
PCL	32.41	62.60	56.85	61.63	38.40
O-PCL	28.53	55.50	55.70	53.56	34.05

**Table 3 polymers-14-02135-t003:** Summary of relative areas (%) of secondary structural features of BSA identified via Gaussian deconvolution of the amide I band of adsorbed samples collected via FTIR.

Samples	β-Structures (%)	α-Helix	Intermolecular β-Sheet (%)	Random Coil (%)
BSA	37.4	51.27	11.86	-
O-PCL	47.36	44.51	7.60	-
PCL	44.87	-	1.27	52.93
Multilayer Scaffold	75.57	20.24	3.90	-

**Table 4 polymers-14-02135-t004:** Langmuir and Freundlich model parameters for fitting the FBS proteins adsorption isotherms. *q_m_* (μg/cm^3^) is the monolayer capacity of the adsorbent, *N* is the constant exponent that describes the intensity of the adsorption process attributed to deviation from linearity, R^2^ is the correlation coefficient of the goodness of fit, and *K_L_* (mL/mg) and *K_f_* are the Langmuir and Freundlich isotherm constants.

Langmuir Model				Freundlich Model			
Isotherm Parameters	PCL	O-PCL	Multilayer Scaffold	Isotherm Parameters	PCL	O-PCL	Multilayer Scaffold
*q_m_* (μg/cm^3^)	11.12	23.98	32.15	N	2.68	5.52	1.79
*K_L_* *	0.15	0.44	0.12	*K_f_* *	2.61	12.2	4.56
*R_L_*	0.93	0.81	0.94	-	-	-	-
*R* ^2^	0.98	0.87	0.95	*R* ^2^	0.85	0.97	0.97

* *K_L,f_* = *K_adsoption_*/*K_desorption_*.

**Table 5 polymers-14-02135-t005:** Pseudo-first-order (PFO) and pseudo-second-order (PSO) kinetic models parameters for FBS protein adsorption on PCL, O-PCL, and the multilayer scaffold. *q_e_* (μg/cm^3^) is the protein adsorbed at equilibrium, R^2^ is the correlation coefficient of the goodness of fit, and *K_L_* (min^−1^) and *K_f_* (min^−1^) are the kinetic constants of PFO and FSO.

Parameters	PFO			PSO		
	PCL	O-PCL	Multilayer Scaffold	PCL	O-PCL	Multilayer Scaffold
*q_e_* (μg/cm^3^)	3.21	18.05	31.84	5.03	25.83	31.84
*K*_1_ (min^−1^)	0.20	0.03	0.03	-	-	-
*K*_2_ (min^−1^)	-	-	-	0.11	0.006	0.031
*R* ^2^	0.93	0.98	0.94	0.99	0.99	0.99
MAE	0.47	0.065	0.53	0.13	0.024	0.009
RMSE	0.69	0.26	0.73	0.35	0.16	0.092

**Table 6 polymers-14-02135-t006:** Tensile properties of pristine polymers and multilayered scaffold before and after protein adsorption.

Mechanical Properties	Before FBS Protein Adsorption			After FBS Protein Adsorption		
	PCL ^a^	PCL-O ^a^	Multilayered Scaffold ^b^	PCL ^a^	PCL-O ^a^	Multilayered Scaffold ^b^
Tensile strength (MPa)	8.59 ± 0.24	7.413 ± 1.34	4.60 ± 0.21	27.97 ± 0.43	25.17 ± 1.01	8.45 ± 1.12
Tensile elongation (%)	250.32 ± 1.7	7.5 ± 0.36	19.42 ± 0.11	250.32 ± 1.7	248.76 ± 2.34	10.89 ± 0.43
Young’s modulus (GPa)	1.43 ± 0.26	1.74 ± 0.97	1.57 ± 0.35	1.56 ± 0.12	2.06 ± 0.32	1.73 ± 0.21

^a^ Quantified using ISO 7198: 2016, ^b^ quantified using ASTM D882.

## Data Availability

Not applicable.

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
