# Peer review of "Blood-Vessel-Inspired Hierarchical Trilayer Scaffolds: PCL/Gelatin-Driven Protein Adsorption and Cellular Interaction"

_polymers, 2022, doi:10.3390/polym14112135_

Round 1

Reviewer 1 Report

This manuscript proposed a method to fabricate three-layered polycaprolactone (PCL)- 27 based tubular structures that contain biochemical cues to improve protein adsorption. PCL was backbone-oxidized and subsequently casted over a photolithography-manufactured microgrooved mold to obtain a bioactive surface.

  1. Surface modification has been widely used to increase biocompatibility, the authors should provide more relevant refences in the introduction.
  2. The fibrous structures fabricated via electrospinning have porous structures, it favors cell adhesion and spreading, however, due to the nano or submicro holes, it is difficult for cell to infuse to the internal area of the structure, and it didn’t give a clear view of the cell penetration in Fig.9, the cells are not evenly distributed as well, therefore, further evaluation is required.

Author Response

Title: Blood-Vessel-Inspired Hierarchical Trilayer Scaffolds: PCL/Gelatin-Driven Protein Adsorption and Cellular interaction

Journal: Polymers (ISSN 2073-4360)

Submission ID: polymers-1710565

Response to reviewers

The original remarks from the reviewer are shown below in bold. Changes in the manuscript are explained herein and introduced in the revised document in the track changes option.

REVIEWER 1 EVALUATION

Comments:

This manuscript proposed a method to fabricate three-layered polycaprolactone (PCL) - 27 based tubular structures that contain biochemical cues to improve protein adsorption. PCL was backbone-oxidized and subsequently casted over a photolithography-manufactured microgrooved mold to obtain a bioactive surface.

  1. Surface modification has been widely used to increase biocompatibility; the authors should provide more relevant references in the introduction.

Answer:

The introduction was reorganized to further show how surface modification have been used to increase biocompatibility, some new references and examples have been included as follows:

Nevertheless, the lack of surface bioactivity of most BSP limits its further applicability to promote cell adhesion, growth, and proliferation [6]. For instance, although different tissue engineered vascular grafts (TEVG’s) made of BSPs, such as PLA – Polylactic acid, PCL – Polycaprolactone, PGA – Polyglycolic acid, PU – Polyurethane, PVA - Polyvinvyl acetate, and even PS (Polystirene), have been reported to be tested on preclinical animal models, none of them have reached clinical trials lacking the required features to over-come the current limitations [11][12].

To address this issue, polymers have been functionalized with CO2 and NO2 plasma to improve hydrophilicity and protein adsorption as well as conjugate bioactive mole-cules[13] such as adhesive peptides ( e.g., CAG, RGD, YIGSR)[14],[15] and antithrombo-genic agents (e.g., PEG, PEI, Heparin) [16],[17],[18]. Other strategies include blends with natural polymers such as gelatin, chitosan, and cellulose [19]. Despite of the successful results in terms of surface bioactivity enhancement, functionalization processes usually involve highly sophisticated and costly procedures and equipment [20] as well as volatile fluorine-toxic solvents (e.g., trifluoroethanol, trifluoracetic acid, hexafluoroisopropanol, among others) to achieve dissolution of blends, which has risen several environmental and health concerns due to their end-of-life products [21].

  1. The fibrous structures fabricated via electrospinning have porous structures, it favors cell adhesion and spreading, however, due to the nano or submicro holes, it is difficult for cell to infuse to the internal area of the structure, and it didn’t give a clear view of the cell penetration in Fig.9, the cells are not evenly distributed as well, therefore, further evaluation is required.

Answer:

Although we understand your concern, in figure 9f is shown the cell nuclei distribution according to the confocal z-stacks. These samples were not subjected to a histology process; rather the graft wall was placed directly on the microscope and cell nuclei were observed up to 100 μm depth and it is possible to observe the cell infiltration and distribution. Figure 9C shows how the fibroblasts are smaller than the microfibers of the graft, and they are completely immersed into the graft wall and they have created philopodia around the fibers. Furthermore, to confirm this data, we have measured the mean horizontal ferret diameter of the pores of the 75:25 and 95:5 layers, which is the smaller diameter of the pores, we have included this data on figure 6j, and the results suggest that the porosity is enough to allow cell infiltration.

Reviewer 2 Report

The manuscript is on the preparation of electrospun PCL/Gelatin nanofiber-based trilayer scaffolds as artificial blood vessels and characterization of their morphological, structural, mechanical and the biological properties. Based on the findings, PCL-based multilayer scaffolds designed in the study proved to be a suitable platform for fibroblast cell attachment and proliferation. The study is well-intended and well-organized.  It contains substantial experimental work and detailed discussion on the subject. A few suggestions for revision are listed below:

-The abstract of the manuscript should be shortened and given more concisely.

-The same comment is also valid for the introduction section. The general information given in the first 5 paragraphs should be concise as possible and further discussion about the previous studies on the same subject should be given in detail.

- The labels of Fig. 4 (g) on the schematic representation of the proposed adsorption mechanism of FBS proteins should be revised (not clear in the current version).

- It is recommended to mention the XRD analysis in the experimental section of the main document as well as the few results given as the supplementary.

Author Response

Title: Blood-Vessel-Inspired Hierarchical Trilayer Scaffolds: PCL/Gelatin-Driven Protein Adsorption and Cellular interaction

Journal: Polymers (ISSN 2073-4360)

Submission ID: polymers-1710565

Response to reviewers

The original remarks from the reviewer are shown below in bold. Changes in the manuscript are explained herein and introduced in the revised document in the track changes option.

REVIEWER 2 EVALUATION

Comments:

The manuscript is on the preparation of electrospun PCL/Gelatin nanofiber-based trilayer scaffolds as artificial blood vessels and characterization of their morphological, structural, mechanical and the biological properties. Based on the findings, PCL-based multilayer scaffolds designed in the study proved to be a suitable platform for fibroblast cell attachment and proliferation. The study is well-intended and well-organized.  It contains substantial experimental work and detailed discussion on the subject. A few suggestions for revision are listed below:

  1. The abstract of the manuscript should be shortened and given more concisely.

Answer:

The abstract was indeed edited to be shortened and concisely explained.  The edited abstract is shown below:

Abstract: Fabrication of scaffolds with hierarchical structures exhibiting the blood vessel topo-logical and biochemical features of the native extracellular matrix that maintain long term pa-tency remains as a major challenge . Within this context, scaffold assembly using biodegradable synthetic polymers (BSP) by electrospinning had led to soft-tissue-resembling microstructures that allow cell infiltration. However, BSP fail to exhibit the sufficient surface reactivity, limiting protein adsorption and/or cell adhesion jeopardizing the overall graft performance. Here we present a methodology for the fabrication of three-layered polycaprolactone (PCL)-based tubu-lar structures with biochemical cues to improve protein adsorption and cell adhesion. For this purpose, PCL was backbone-oxidized (O-PCL) and casted over a photolithogra-phy-manufactured microgrooved mold to obtain a bioactive surface as demonstrated by a pro-tein adsorption assay (BSA), Fourier transform infrared spectroscopy (FTIR) and calorimetric analyses. Then, two layers of PCL:Gelatin (75:25 and 95:5 w/w), obtained by a novel single-desolvation method, were electrospun over the casted O-PCL to mimic a vascular wall with a physicochemical gradient to guide cell adhesion. Furthermore, tensile properties showed to withstand the physiological mechanical stresses and strains. In vitro characterization, using L929 mouse fibroblasts, demonstrated that the multilayered scaffold is a suitable platform for cell in-filtration and proliferation from the innermost to the outermost layer as is needed for vascular wall regeneration. Our work holds a promise as a strategy for the low-cost manufacture of next-generation polymer-based hierarchical scaffolds with high bioactivity and resemblance of ECM’s microstructure to accurately guide cell attachment and proliferation.

  1. The same comment is also valid for the introduction section. The general information given in the first 5 paragraphs should be concise as possible and further discussion about the previous studies on the same subject should be given in detail.

Answer:

The Introduction was modified to resume the need statement as follows:

Cardiovascular diseases represent one of the main causes of death in the world. About 8.5 million people worldwide have diseased blood vessels that must be replaced with vascular grafts [1],[2]. Currently, commercially available synthetic vascular grafts are made of several non-biodegradable materials such as Polytetrafluoroethylene (PTFE) and Polyethylene terephthalate (PET)[3]. However, their performance is inefficient in replacing small diameter vessels (< 6mm) due to their limited long-term patency with only 32% success after 2 years [4],[5] caused by thrombogenic events [3], intimal hyperplasia[6], [7] and/or atherosclerosis [8], triggering several acute or chronic inflammatory responses [6].

The introduction was reorganized and a paragraph was added to deepen on the previous studies showing the state of the current strategies and their success rates, in this way supporting the idea of our approach being a novel strategy to recover the appropriate function of the native tissues with hierarchical structures as shown:

These complex interactions explain the reason of preclinical models on TEVG’s showing that near of the 14% of them fail in early stages either on the peri-implantation stages or within the first months, and 35% of them report patency loss or need of removal before the beginning of the tissue remodeling. In this sense, although the graft have been successfully implanted, the substrate is not suitable for tissue remodeling either due to thrombogenesis, generation of exacerbated acute inflammatory responses, or due to lack of regeneration caused, in both cases, due to limited cell infiltration and interaction given by the BSP properties and scaffold microstructure [4].

  1. The labels of Fig. 4 (g) on the schematic representation of the proposed adsorption mechanism of FBS proteins should be revised (not clear in the current version).

Answer:

The image was modified and labels were revised to make them clear.

  1. It is recommended to mention the XRD analysis in the experimental section of the main document as well as the few results given as the supplementary.

Answer:

The method was included as follows, indicating the complete description on Appendix A2.

PCL and O-PCL ray diffraction analysis (XRD) were taken in a Rigaku Ultima III X-ray diffractomer (Tokio, Japan) in the Bragg-Brentano configuration. Samples with an average contact area of 25 mm2 were placed into a bounding grid of 5 mm and exposed to a Bragg difraction angle (2θ) sweep between 5° and 50° with a 1.5°/seg step using Kα radiation with a Copper (Cu) anode of 40 kV and 40 mA. The Debye-Sherrer equation was used to estimate the apparent crystal size of the polymeric structure based on the full-width at half-maximum of the x-ray diffraction line, also known as FWHM, the wavelength of the X-ray used which was 1.5406 nm and the angle between the incident ray and the scattering planes. The phase content of each sample was calculated from the XRD profiles by peak deconvolution using the Materials Data Jade 9® Software (Newtown Square, PA, USA) using Gaussian profiles as fitting methods (Appendix A2).

A new section was added with the results from the X-Ray Diffraction indicating the complete data on Appendix B3.

3.1.3. X-Ray Diffraction

XRD pattern of PCL and O-PCL films at room temperature displays three strong diffrac-tion peaks at Bragg angles 2θ= 21.4°, 22° and 23.8° that correspond to the orthorhombic planes (110), (111) and (200) of the crystalline phase of PCL. Also, a halo centered near 21° in the XRD profile indicates the presence of amorphous structures in the samples as ex-pected from the semicrystalline structure of PCL. Observable differences can be identified in the deconvoluted peak intensities and peak widths among the orthorhombic unit cells diffractions when oxidation takes place, indicating variations on the lamellar organiza-tion of the polymer.

Nevertheless, since there were no additional peaks on the XRD pattern of the O-PCL, it can be inferred that the oxidation process did not significantly affect the semicrystalline structure and thus, only appears to be a surface phenomenon [3]. Finally, the XRD pattern exhibits that are comparable with those obtained in previous studies where OCFG on the backbone of PCL was generated with CO2 plasma treatment, indicating that the oxidation introduced here is a cost-effective alternative for polyester oxidation (Appendix B3).

Round 2

Reviewer 1 Report

the manuscript can be accepted, the revision is satisfying.